# Abdominal Compartment Syndrome in Acute Pancreatitis: A Narrative Review

**DOI:** 10.3390/diagnostics13010001

**Published:** 2022-12-20

**Authors:** Narcis Octavian Zarnescu, Ioana Dumitrascu, Eugenia Claudia Zarnescu, Radu Costea

**Affiliations:** 1Department of General Surgery, “Carol Davila” University of Medicine and Pharmacy, 050474 Bucharest, Romania; 2Second Department of Surgery, University Emergency Hospital Bucharest, 050098 Bucharest, Romania

**Keywords:** acute pancreatitis, abdominal compartment syndrome, compartment syndrome, intra-abdominal hypertension, surgery

## Abstract

Abdominal compartment syndrome (ACS) represents a severe complication of acute pancreatitis (AP), resulting from an acute and sustained increase in abdominal pressure >20 mmHg, in association with new organ dysfunction. The harmful effect of high intra-abdominal pressure on regional and global perfusion results in significant multiple organ failure and is associated with increased morbidity and mortality. There are several deleterious consequences of elevated intra-abdominal pressure on end-organ function, including respiratory, cardiovascular, gastrointestinal, neurologic, and renal effects. It is estimated that about 15% of patients with severe AP develop intra-abdominal hypertension or ACS, with a mortality rate around 50%. The treatment of abdominal compartment syndrome in acute pancreatitis begins with medical intervention and percutaneous drainage, where possible. Abdominal compartment syndrome unresponsive to conservatory treatment requires immediate surgical decompression, along with vacuum-assisted closure therapy techniques, followed by early abdominal fascia closure.

## 1. Introduction

Severe acute pancreatitis (AP) is defined by the presence of persisting organ failure (>48 h) and represents a small subgroup (less than 5–10%) of acute pancreatitis patients [1,2]. Severe AP has a high mortality rate, approximating 35%, and is characterized by cytokine activation, systemic inflammatory syndrome (SIRS), and organ failure (with significant importance of duration, type, and number) [3,4,5]. Organ-dysfunction assessment is based on the SOFA (Sequential Organ Failure Assessment) score, requiring a score higher than three [6]. IAH (intra-abdominal hypertension) is defined as intra-abdominal pressure values over 12 mmHg, obtained after two measurements at intervals of 1–6 h [7]. Abdominal compartment syndrome (ACS) is defined by a sustained increase in intra-abdominal pressure above 20 mmHg that is associated with the appearance of new organ dysfunction, further increasing morbidity and mortality [7]. Acute pancreatitis represents a risk factor for intra-abdominal hypertension (IAH) and ACS, with an incidence of 50–60% for IAH and 15–30% for abdominal compartment syndrome [8,9,10].

The mortality rate for ACS in severe acute pancreatitis is between 25% and 83% [11,12]. The development of IAH in acute pancreatitis brings a poor prognosis that is reflected by a higher (i.e., worse) prognostic score (APACHE II), higher rates of MSOF (multiple system organ failure), and increased mortality [13,14,15]. Intra-abdominal hypertension and ACS are common entities that often remain unrecognized or underdiagnosed [16]. Understanding the etiology and pathophysiology of IAH and ACS is essential for early diagnosis and underpins prevention and therapy. The purpose of this paper is to review the current evidence on pathophysiology, diagnosis, and therapeutic management of ACS in acute pancreatitis.

## 2. Pathophysiology

The abdominal cavity has walls that are partially rigid (spine, pelvis, and ribs) and a portion that is mostly flexible (abdominal wall and diaphragm), but it thus has limited compliance. Intra-abdominal pressure is determined by the flexibility of the abdominal wall and the volume of abdominal contents. The ability to expand the abdomen is measured by abdominal compliance (AC), which is dependent on the elasticity of the abdomen and diaphragm. Compliance can be represented as the change in intra-abdominal volume per change in intra-abdominal pressure. Abdominal compliance is dependent on both of these components; therefore, evaluating its variations over time is difficult [17,18]. Particularly, in acute pancreatitis, the abdominal wall compliance is reduced by abdominal pain and abdominal wall edema [19]. Abdominal compliance can only be measured when there is a change in intra-abdominal volume, such as when intra-abdominal free fluid draining is performed. Normal abdominal compliance is around 250–450 mL/mmHg [20]. A minor decrease in intra-abdominal volume may result in a large decrease in intra-abdominal pressure (IAP) in individuals with ACS and diminished abdominal compliance [17,21]. Male gender, android body composition, fluid overload, pseudocyst, abscess, and prone stance are related to reduced abdominal compliance. In fact, most individuals who are described in research as having ACS linked with acute pancreatitis (Table 1) are male. The treatment of patients with diminished AC and IAH is founded on the same principles [20,22,23]. The model of interactions between multiple compartments is acknowledged as the polycompartment model [24,25]. Changes in thoracic compliance, for example, will be reflected in abdominal compliance and vice versa [26]. As a result, increased IAP will decrease chest wall compliance.

Intra-abdominal pressure is typically between 5 and 7 mmHg in healthy individuals and 10 mmHg in seriously ill adults [27]. Any change in the volume of one intra-abdominal compartment affects the other compartments, resulting in alterations in abdominal perfusion pressure and intra-abdominal pressure. In the setting of acute pancreatitis (Figure 1), both the retroperitoneum (due to an enlarged pancreas and fluid collections) and the peritoneal cavity might expand (due to ileus, ascites, and bowel edema). Sequestration of up to six liters of fluid has been described in severe acute pancreatitis 48 h after onset. In the abdominal cavity, volume fluctuations are first corrected so that the intra-abdominal pressure remains nearly constant up to a crucial threshold. If this critical volume is exceeded, intra-abdominal pressure builds up rapidly, leading to intra-abdominal hypertension and abdominal compartment syndrome [28]. In particular, microcirculation abnormalities have the most significant impact on the incidence of IAH and ACS in acute pancreatitis. Several pro-inflammatory cytokines and vasoactive mediators are recruited, resulting in increased capillary permeability, fluid extravasation, and hypovolemia. Fluid buildup begins in the retroperitoneal and interstitial space of the gastrointestinal tract [29]. Retroperitoneal edema, intestinal dysfunction (ileus), SIRS, a continuous increase in capillary permeability, and abdominal wall rigidity brought on by pain result in vascular compression, visceral compression, decreased venous flow, cellular hypoxia, increased interstitial edema, and increased intra-abdominal pressure [30].

The importance of aggressive volume resuscitation and a positive fluid balance in the onset of IAH and ACS is correlated with a positive fluid balance. The optimal fluid resuscitation therapy should both overcome systemic hypovolemia caused by intravascular fluid loss and prevent (or reduce) the accumulation of body fluid in retroperitoneal spaces [31]. In patients with acute pancreatitis, local microcirculatory disruption leads to hypoperfusion, and ischemia of pancreatic tissue has been documented [32,33]. Therefore, the objective of volume resuscitation is to maintain adequate vascular volume, while also enhancing tissue oxygenation and microcirculation. Indicators of a favorable response to volume resuscitation include a central venous pressure between 8 and 12 mmHg, a mean arterial pressure greater than or equal to 65 mmHg, a urinary volume greater than or equal to 0.5 mL/kg/h, and an oxygen saturation greater than or equal to 0.80 [34]. If any of these parameters indicate an imminent volume overload, aggressive fluid administration should be avoided due to the risk of increased “third space” and intra-abdominal pressure, which can lead to the development of abdominal compartment syndrome [35]. There are several deleterious consequences of elevated IAP on end-organ function, including renal, respiratory, cardiovascular, gastrointestinal, and neurologic effects.

### 2.1. Effects on the Reno-Urinary System

The earliest important manifestation of increasing IAP even at relatively low-level of intra-abdominal pressure is oliguria and acute kidney injury. Acute kidney injury (AKI) is one of the most prevalent consequences of AP (70% of patients with acute pancreatitis develop AKI) [36]. The primary causes are the release of pro-inflammatory toxins and renal hypoperfusion, which causes a decrease in glomerular filtration rate, oliguria, or even anuria. Increased intra-abdominal pressure in AP exacerbates AKI via prerenal and renal processes [37]. Due to diminished cardiac function and cardiac output, the prerenal mechanism involves a decrease in renal perfusion pressure [38]. The renal mechanism is represented by the increase in renal vascular resistance, due to renal extrinsic compression [38,39]. It was estimated that, at an IAP of 20 mmHg, renal resistance increases by 500%, and at an IAP of 40 mmHg, it can increase up to 1500% [17,40].

These disruptions result in a decrease in the rate of glomerular filtration and the production of renin, antidiuretic hormone, and aldosterone, leading to a general increase in vascular resistance and thus sustaining a vicious circle. These events lead to a decrease in urine flow, with oliguria occurring at an IAP of 15–20 mm Hg and anuria at an IAP of 30 mm Hg [40]. These alterations shunt blood from the kidney, resulting in glomerular necrosis, tubular damage, and the progression of renal insufficiency [41].

### 2.2. Effects on the Gastrointestinal System

Several investigations have demonstrated that intestinal vascular anomalies may exacerbate the severity of AP during its progression [42,43]. This facilitates the modification of the intestinal barrier and the translocation of bacteria from the colon to the portal venous system and lymphatic system, hence triggering SIRS and MODS [44,45,46]. The digestive tract is extremely sensitive to elevated IAP levels. A rise in IAP to 40 mmHg can reduce blood flow to the mesenteric artery by 69%, leading to bowel ischemia [47]. Compression of the mesenteric veins results in intestinal edema, with the continuous increase in IAP and closure of a vicious circle, leading to decreased intestinal perfusion, decreased intraluminal pH, intestinal ischemia, metabolic acidosis, and increased mortality [47].

Hepatic venous, arterial, and microcirculatory blood flow decreases significantly with even slight increases in intra-abdominal pressure. Hepatic dysfunction may lead to decreased clearance of plasma lactate, further adding to metabolic acidosis [9,48]. Hepatorenal syndrome is one of the major consequences of end-stage liver disease. The vasodilation of the periphery is recognized as the classic pathogenic process in the development of hepatorenal syndrome, which is accompanied by an increase in intra-abdominal pressure. The defining characteristic of hepatorenal syndrome is an early onset of severe renal vasoconstriction. In mice models, a study investigated the potential role of IAH in the development of hepatorenal syndrome [49]. For the IAP = 10 cm H2O, significant constrictive renal tubular lumen with significant inflammatory infiltration in the renal interstitial and cellular swelling were detected. Moreover, the formed casts and hyperemia in the renal interstitial and the edema of renal tubular epithelial cells were the characteristics of IAP = 20 cm H2O. The authors concluded that IAH was the significant pathological mechanism and an independent risk factor in the occurrence and development of hepatorenal syndrome [49].

### 2.3. Effects on the Respiratory System

Lungs are the primary target of proinflammatory mediators during acute pancreatitis, resulting in increased alveolo-capillary permeability, decreased surfactant levels, and decreased pulmonary perfusion [50]. Higher IAP exerts a direct mechanical effect on the lung, leading to decreased thoracic volume and increased intrathoracic pressure [51]. Passive ascension of the diaphragm permits the transmission of intra-abdominal pressure into the pleural cavity, hence decreasing bilateral static and dynamic lung compliance [52].

Through the raising of the diaphragm, there is an increase in intrathoracic pressure, resulting in extrinsic compression of the lung parenchyma with the onset of alveolar atelectasis, decreased diffusion of oxygen and carbon dioxide through the alveolo-capillary membrane, and an increase in intrapulmonary shunt fraction and alveolar dead space. The presence of hypovolemia aggravates these dysfunctions [53]. Thus, the increase in pCO2 results from a decrease in pulmonary compliance and ineffective breathing. Additionally, pO2 is lowered due to basal atelectasis and decreased cardiac output. Under these circumstances, hypercapnia and acidosis may result in catastrophic respiratory failure [54].

### 2.4. Effects on the Cardiovascular System

Interstitial edema and cardiomyocyte hypoxia, myofiber over-contractility, intercellular edema between the cardiomyocytes, and cardiomyocyte hypertrophy with collagenization of the myocardial stroma are the main ultrastructural changes that occur at the myocardial level during acute pancreatitis [55]. In addition, electrolyte disorders associated with AP have a significant impact on the cardiovascular system (variations in calcium concentration, hypophosphatemia, and hyperkalemia) [47]. During AP, the effects of IAP on the hemodynamic system are complex. This is primarily due to the pressure exerted on the large blood vessels and heart. Consequently, there is a decrease in venous return, an increase in pulmonary artery pressure and central venous pressure, a decrease in cardiac output, an increase in peripheral vascular resistance, and favorable conditions for the onset of venous thrombosis or pulmonary embolism.

The three essential components of cardiac function are preload, contractility, and afterload. Elevated IAP has a detrimental effect on each of these interdependent components, and restoring each to an adequate level is crucial for improving abdominal and systemic perfusion, oxygen transport, and patient outcome [56]. Preload is reduced because of fluid sequestration in the splanchnic area, vascular bed of the lower extremity, and the inferior vena cava. In addition, increasing intra-abdominal pressure reduces compliance of the left ventricle and thus decreases ventricular filling. Progressive impairment of cardiac output leads to cardiovascular collapse and shock as the terminal events.

### 2.5. Effects on the Central Nervous System

Intra-abdominal hypertension directly affects cerebral perfusion and central nervous system function through the elevation of intracranial pressure. Elevations in intra-abdominal and intrathoracic pressure may also directly impact the pressures within the cranium [57]. Proposed mechanisms have included decreased lumbar venous plexus blood flow; increased pCO2 level, resulting in increased cerebral blood flow; and decreased cerebral venous outflow [58,59]. Intracerebral venous pooling can markedly worsen pre-existing cerebral perfusion abnormalities due to trauma, chronic intracranial hypertension, or other causes of decreased cerebral compliance.

## 3. Classification of Intra-Abdominal Hypertension

The following degrees of severity of IAH have been proposed to stratify risk and guide therapy [26]:Grade I: intra-abdominal pressure: 12–15 mmHg.Grade II: intra-abdominal pressure: 16–20 mmHg.Grade III: intra-abdominal pressure: 21–25 mmHg.Grade IV: intra-abdominal pressure: >25 mmHg.

Grades I and II can be treated conservatively, but grades III and IV require surgery. Another classification of the IAH may be based on the duration of symptoms [26]:Hyperacute: IAP increases in the order of seconds-minutes that occur in certain situations: laughter, coughing, sneezing, defecation.Acute: IAH lasting several hours in the trauma surgical patient or intra-abdominal bleeding; this entity can evolve fulminating in a few hours to the abdominal compartment syndrome.Subacute: IAH that appears progressively over days, frequently found in patients with severe acute pancreatitis. The typical example being patients with medical pathology hospitalized in the intensive care unit (massive resuscitation in the patient with severe burns, and leakage capillary syndrome associated with sepsis).Chronic: IAH that develops progressively in months or years in the context of pregnancy, morbid obesity, peritoneal dialysis, and liver cirrhosis with ascites); these patients are at risk of developing acute intra-abdominal hypertension in the event of a critical illness.

## 4. Classification of Abdominal Compartment Syndrome

Abdominal compartment syndrome is now recognized as a cause of significant organ failure, morbidity, and mortality in all critically ill patients [17,60]. Considering the multitude of predisposing conditions that may lead to the development of IAH/ACS, the consensus in 2006 classifies ACS as either primary, secondary, or recurrent according to the duration and cause of the patient’s IAH [26].

Primary ACS (surgical/postoperative/abdominal) is characterized by an acute/subacute increase in intra-abdominal pressure in certain circumstances: abdominal trauma, abdominal aneurysm dissection, hemoperitoneum, acute pancreatitis, secondary peritonitis, retroperitoneal hemorrhage, and liver transplantation; it frequently requires surgery early or radiological interventional therapy.Secondary ACS (medical/extra-abdominal) is characterized by a subacute/chronic increase in intra-abdominal pressure that occurs secondary to extra-abdominal causes: sepsis, capillary leakage, severe burns, or other conditions that require massive resuscitation.Recurrent ACS (tertiary) represents the reappearance of abdominal compartment syndrome after resolution of a previous episode of primary or secondary abdominal compartment syndrome; it is associated with acute intra-abdominal hypertension, being equivalent to a “second-hit”, having morbidity and significantly increased mortality.

According to the 2006 consensus regarding the classification of ACS, we conclude that, regarding severe acute pancreatitis, we can talk about a primary ACS.

## 5. Diagnosis

### 5.1. Clinical Diagnosis

It is crucial to diagnose abdominal compartment syndrome quickly and begin treatment without delay to prevent irreversible tissue damage. Abdominal compartment syndrome is a serious complication that is observed in ICU (intensive care unit) patients with severe acute pancreatitis whose condition is deteriorating. It has been reported that a physical examination is inaccurate for IAH diagnosis [61,62]; consequently, precise IAP measurement is important. In patients with neurogenic bladder or benign prostatic hypertrophy, the use of bladder pressures is contraindicated. Intragastric measurement may be a viable alternative to intravesical measurement, allowing for continuous monitoring [63,64]. When measuring IAP intermittently, WSACS (https://www.wsacs.org) recommends using the bladder, with a maximum instillation volume of 25 mL of sterile saline, followed by measurement at end-expiration in the complete supine position after ensuring that abdominal muscle contractions are absent and with the transducer zeroed at the level of the mid-axillary line. If the patient is not sedated or supine, bladder pressure measurements may be erroneous. More recent methods of IAP measurement include wall thickness (using point-of-care ultrasound, POCUS), wireless motility capsule, continuous IAP measurement device, and IAP estimation using near-infrared spectroscopy [65,66]. IAP should be expressed in mmHg. IAH is defined as intra-abdominal pressure values over 12 mmHg, obtained after two measurements at intervals of 1–6 h [7].

### 5.2. Imaging Diagnosis

Several studies have highlighted the role of POCUS in the diagnosis and management of IAH [54,67]. Thus, ultrasound evaluates the position of the nasogastric tube (NG-tube) in the stomach, gastric contents, intestinal motor function and intestinal contents, and the presence of ascites. POCUS should be performed at 6-hour intervals immediately after IAP measurement. Limitations are related to the measurement of the width of the inferior vena cava in obese people, where there is already a certain degree of compression, which can be misinterpreted as hypovolemia. From a therapeutic point of view, ultrasound-guided drainage of intra-abdominal and peripancreatic fluid collections seems safe and effective [68,69].

A computed tomography exam of the abdomen is essential for the diagnosis and grading [1], and it is effective for the accurate prediction of the progression of acute pancreatitis [70,71]. In addition, Gupta et al. found that computed tomography findings such as the presence of pancreatic necrosis, abdominal collections, ascites, pleural effusions, intestinal-wall thickening (>3 mm), bowel dilation, bowel-wall enhancement, and biliary dilatation are valuable signs for the diagnosis of intra-abdominal hypertension [72]. The presence of collection, volume and maximum dimension of the fluid collections, biliary dilatation, and presence of moderate pleural effusion were CT features that were significantly associated with the presence of IAH in severe acute pancreatitis. Another study showed that the presence of a round-belly sign, moderate–gross ascites, and pancreatic necrosis of >50% on contrast-enhanced computed tomography can predict presence of IAH in acute pancreatitis [73]. The round-belly sign (Figure 2) is considered positive when the ratio of anteroposterior-to-transverse (AP–T) abdominal diameter measured at the level where the left renal vein crosses the aorta, not including the subcutaneous fat, is greater than 0.8 [74].

Other studies mention other radiological signs for IAH, such as elements of increased intra-luminal contents (gastric or bowel distention), increased intra-abdominal contents (hemoperitoneum/active abdominal bleeding, intra-peritoneal fluid collections, intra-abdominal free air), elevation of the diaphragm, narrowing of the intrahepatic portion of inferior vena cava (narrowing <3 mm), contour deformity, visceral compression, or bilateral inguinal hernia [75,76].

### 5.3. Laboratory Diagnosis

An evaluation of serum or urinary biomarkers was performed both on experimental and human subjects to assess the possibility of an earlier diagnosis of intra-abdominal hypertension or abdominal compartment syndrome. Several biomarkers were examined: kidney dysfunction (BUN and creatinine), intestinal damage (D-lactate [77,78], D-dimer [79], Intestinal Fatty Acid Binding Protein, and I-FABP [80,81]), aspartate aminotransferase [80], oxidative stress biomarkers such as glutathione [82,83], superoxide dismutase isoenzymes [84], and fatty acid ethyl esters [85].

Due to the fact that the cytokine cascade from the innate immune system and the activated adaptive immune system (including CD4+ and CD8+ T lymphocytes) are critical for the development of SIRS in acute pancreatitis, the role of T lymphocytes and inflammatory cytokines in abdominal compartment syndrome in severe acute pancreatitis were investigated [86,87]. A retrospective analysis of 76 patients with severe acute pancreatitis (36 patients with ACS and 40 with intra-abdominal hypertension) revealed that proportions of CD4+ T lymphocytes on days 1, 3, and 7 were significantly lower in ACS patients than in IAH patients, whereas proportions of CD8+ T cells did not differ significantly between the two groups on any of the three days [88]. In the same study, ACS patients experienced a substantial decrease in the CD4+/CD8+ ratio on day 1, but not on days 3 and 7. A prospective study of twenty-five ACS patients with SAP revealed a substantial positive connection between IL-8 blood levels and intra-abdominal pressure [89]. This study showed that early continuous veno-venous hemofiltration reduces IAP and IL-8 levels in the blood of ACS patients with severe acute pancreatitis. In an examination of 25 surgical patients with IAP ≤12 mmHg and 45 surgical patients with IAP >12 mmHg, Bodnar et al. identified a correlation between IAP and serum adenosine and IL-10 levels [90].

## 6. Treatment

### 6.1. Non-Surgical Management

Patients with severe AP require permanent monitoring of intra-abdominal pressure and evaluation of organ function for rapid diagnosis of ACS and prompt initiation of treatment [7,21,91]. Once the diagnosis has been made, treatment should be instituted as soon as possible. Due to the consequences of a clinically significant increase in measured intra-abdominal pressure, head-of-bed elevation should be avoided [92]. WSACS recommends a treatment algorithm that targets ACS at several key points. The first stage of treatment in the management of ACS involves non-surgical measures that can be definitive in some patients [54,93]. 

#### 6.1.1. Evacuate Intraluminal Contents

The first step is to evacuate the intestinal contents by inserting the nasogastric tube and the rectal drainage tube and then employ the use of prokinetics, thus minimizing enteral nutrition and administration of enemas [65]. Neostigmine can be used to increase intestinal peristalsis, and it was proposed that it be used to treat colonic ileus associated with IAH that does not respond to other basic treatments [18]. There was a recent single-center randomized trial that compared intramuscular neostigmine (1 mg every 12 h increased to every 8 h or every 6 h, depending on response) and continued conventional treatment for 7 days for patients with intra-abdominal hypertension in acute pancreatitis [94]. The authors found that neostigmine was considerably more effective than conventional treatment in lowering IAP in AP patients with persistent IAH after 24 h of conventional treatment. When intra-abdominal pressure surpasses 12 mmHg, erythromycin and metoclopramide are advised as further prokinetic agents [8]. In nonresponsive patients, endoscopic decompression of the colon may be an option. 

#### 6.1.2. Improve Intra-Abdominal Compliance

The third step involves increasing compliance of the abdominal wall by using neuromuscular blockers, which are especially useful in patients who cannot benefit from surgical decompression [95]. A Belgium study that included ten patients showed that bolus administration of cisatracurium at a dose of 0.15 mg/kg can be used to temporarily reduce IAP in patients with intra-abdominal hypertension [96]. One report described successful treatment of ACS with prolonged neuromuscular blockade with atracurium, avoiding a laparostomy [97]. An important role in this stage is represented by an analgesia and an adequate sedation. The elimination of possible abdominal eschars can also be considered. By lowering pain, agitation, and accessory muscle use, these interventions can increase thoracoabdominal muscular tone and abdominal-wall compliance [65].

#### 6.1.3. Optimize Fluid Administration and Improve Systemic/Regional Perfusion

A central element of the prevention of IAH and ACS in severe AP is the adequate volume resuscitation. A critical step of acute pancreatitis treatment is to obtain an optimal systemic infusion through the correct administration of fluids. For acute pancreatitis, several guidelines recommend quickly supplementing with isotonic crystalloid solution to restore end-organ perfusion [98,99,100,101]. Volume resuscitation should be adjusted according to the patient’s responsiveness to it, the time elapsed since the onset of pancreatitis (the first 24 h are crucial), and the patient’s propensity for fluid sequestration. At this step, the intensive-care physician has an important role in correcting hypovolemia and preventing the iatrogenic occurrence of abdominal compartment syndrome. The fluid needs of critically sick patients tend to fluctuate during the course of their illness, and fluid therapy should be adjusted accordingly [102]. Malbrein’s group consequently proposed the ROSE concept to aid in therapeutic decision-making by separating four phases of fluid administration: the resuscitation phase, the optimization phase, the stabilization phase, and the evacuation phase [103]. 

After initial enthusiasm towards active fluid treatment, it shortly became clear that aggressive resuscitation for more than 48 h after the onset of pancreatitis leads to increased mortality through the development of abdominal compartment syndrome [104,105,106]. In a very recent randomized trial involving patients with acute pancreatitis, in the interim analysis of 249 patients, early aggressive fluid resuscitation (a bolus of 20 mL per kilogram of body weight, followed by 3 mL per kilogram per hour) resulted in a higher incidence of fluid overload without improvement in clinical outcomes compared to moderate fluid resuscitation [107]. Because of the risk of under-resuscitation when a fixed infusion rate is utilized and the potential for damage when fluid treatment is administered too aggressively, a more individualized strategy is required [108]. The use of loop diuretics and hemodialysis should be considered as well [55]. 

#### 6.1.4. Antibiotics

Patients with necrotic acute pancreatitis are susceptible to developing infections due to bacterial translocation and weakened immune systems in the early stages and can develop infected pancreatic necrosis about 2–4 weeks after onset [109]. The prophylactic use of antibiotics is not recommended, but the empirical use of antibiotics is recommended in patients who develop organ dysfunctions during severe AP, due to the risk of bacteremia [98,99,101,110]. Despite recommendations, antibiotics have been documented to be used excessively and inappropriately [111,112]. The misuse of antibiotics increases the likelihood of hospital-acquired illnesses. At the same time, early elevated inflammatory markers identify the subset of individuals who gain the greatest benefit from prompt antibiotic therapy [113,114]. However, patients who progress to IAH are at risk for developing sepsis, thus necessitating the administration of antibiotics [8,86].

#### 6.1.5. Energy Nutrition

It is recommended that it not be delayed for more than 48 h in order not to lose its benefits related to bacterial translocation, systemic infections, organ dysfunction, and mortality. It is recommended that general nutrition be given through the naso-jejunal tube, with a volume of 10 mL/h [109]. Enteral nutrition reduces the likelihood of bacterial overgrowth and translocation, hence preventing the development of intra-abdominal hypertension [115,116]. Oral nutrition should be discontinued, and total parenteral nutrition should be commenced in patients with manifest abdominal compartment syndrome. 

### 6.2. Percutaneous Drainage

Intraperitoneal fluid accumulation can be seen frequently in the evolution of acute pancreatitis, but it can also be the result of aggressive volume resuscitation. In the diagnosis of these collections, an important role is played by abdominal ultrasound and abdominal computed tomography, which can also guide the subsequent percutaneous evacuation or paracentesis. The first stage in ACS decompression is percutaneous drainage of the collections via catheter insertion under radiological monitoring. Sun et al. found a link between abdomen pressure and the drainage volume, hospitalization length, and APACHE II score in a trial comparing conservative therapy versus percutaneous drainage for ACS in acute pancreatitis [43]. Thus, individuals who benefited from percutaneous draining had a lower mortality rate (from 20.7% to 10%) than those who received conservative treatment. Another study compared percutaneous drainage (212 patients) with open decompression (61 patients) in the treatment of ACS in patients with early stage found that percutaneous drainage may offer significant benefits for patients when compared with traditional open decompression [117]. However, if percutaneous drainage fails to reduce intra-abdominal pressure, decompressive laparotomy must be performed, and the procedure’s delay can contribute to an increase in mortality [118].

### 6.3. Surgical Treatment

Although a single threshold of IAP cannot be globally applied to decision-making for all patients, progressive organ dysfunction and ACS refractory to medical and percutaneous interventions warrant a prompt surgical decompression [13]. Due to numerous factors, the mortality rate among patients who benefit from decompressive laparotomy remains high. These patients are severely ill at the time of surgery. Second, there is a subset of patients who do not respond to abdominal decompression, with IAH values remaining elevated following decompression. Thirdly, decompression can be detrimental to the patient. Morris et al. described the lethal reperfusion syndrome with the onset of hemorrhage and hemorrhagic shock in patients whose coagulation had not been restored preoperatively [119]. 

There is currently no consensus on the timing or best technique for ACS decompression in patients with severe acute pancreatitis (Table 1). The timing of decompression operation is a subject of dispute. An experimental investigation has demonstrated that both too early and too late decompression should be avoided due to significant morbidity and negative outcomes, respectively [120]. By undergoing reversible surgical decompression that might break the vicious cycle of ACS, there is evidence that early decompression may enhance survival [9,19,40]. Irreversible intestinal ischemia diagnosed too late is a significant factor in the failure of the conservatory strategy [9,121]. Several authors recommended the decompression to be performed between several hours and four days of the diagnosis of abdominal compartment syndrome (Table 1). Due to the severity of the prognosis and the unavailability of alternative treatments, decompensated respiratory or cardiac failure demands immediate surgical decompression [122].

**Table 1 diagnostics-13-00001-t001:** Abdominal compartment syndrome in the context of severe acute pancreatitis—prognosis and surgical management.

First Author (Year)	Severe AP	IAH	IAH—Male (%)	ACS	Interventions	% Interventional Treatment of ACS	Time to Intervention	ACS Mortality
Tao (2003) [123]	345	2	14 (67%)	21	Midline laparotomy with Bogota bag (*n* = 18)	85.7	9–22 h	33.30%
De Waele (2005) [124]	44	21	15 (71%)	4	Midline laparotomy, temporary abdominal closure system (*n* = 4)	100%	-	75%
Chen (2008) [8]	74	44	23 (52%)	20	Percutaneous abdominal decompression and drainage (*n* = 8); Decompressive emergency laparotomy (*n* = 8)	65%	26–33 h	75%
Mentula (2010) [28]	26	0	23 (88%)	26	Open abdomen (*n* = 21) Subcutaneous linea alba fasciotomy (*n* = 5)	100%	1–5 days	46%
Bezmarevic (2012) [12]	51	27	23 (79%)	6	Midline laparotomy (*n* = 6)	83%	1–4 days	83%
Davis (2013) [11]	43	16	16 (100%)	16	Midline laparotomy with Bogota bag (*n* = 11) or wound VAC system (*n* = 5)	100%	3 h	25%
Peng (2016) [117]	273	273	168 (62%)	273	Midline laparotomy (*n* = 61) Percutaneous catheter drainage (*n* = 212)	23.30%	2–101 h	52.50%
Smit (2016) [9]	59	29	21 (72%)	13	Transverse subcostal laparotomy (*n* = 7), midline laparotomy (*n* = 3)	10 (77%)	1.9–15.5 days	53%

The most common technique for abdominal decompression is the median xipho-pubic laparotomy, which permits a thorough exploration of the abdomen [34,125,126]. A further method of decompression is the bilateral subcostal transverse incision, which allows for a quicker primary closure and an easier access to the pancreatic region if subsequent pancreatic surgery is anticipated [40]. 

Alternative to xipho-pubic laparotomy is subcutaneous linea alba fasciotomy (SLAF), which is as minimally invasive treatment method for abdominal compartment syndrome [127]. The incision of linea alba without incising the peritoneum avoids contamination of the peritoneal cavity and permits conversion to median laparostomy [128,129]. This procedure can be aided laparoscopically for enhanced visual control [120]. A review of ten patients that received SLAF reported a 40% mortality rate [130]. Minimally invasive decompression with the assistance of a laparoscope was also proved to be effective in a report of three patients with severe acute pancreatitis associated with abdominal compartment syndrome [131]. The authors’ technique involved a 6 cm long oblique incision from the root of the 12th rib and lateral of the erector spinae that was made in the left or both sides, along with laparoscopic retroperitoneal exploration and necrosectomy.

When the abdominal cavity is opened, intra-abdominal pressure rapidly decreases [132]. It is required to restrict the extent of the operation to findings that may result in imminent mortality. Consequently, during the laparotomy, the surgeon must assess the viability of the intestines and, if necessary, execute an enterectomy, control sources of bleeding, evacuate any septic collections, and obtain swabs for bacterial and fungal culture [21,122,133].

The primary objective following decompressive laparotomy is fascia management. Temporary abdominal wall closure (TAC) can be accomplished by using a variety of techniques, including the Bogota bag, Marlex zipper, Velcro adhesive sheets, absorbable and non-absorbable mesh, and sandwich technique; however, the gold standard is considered to be the vacuum-assisted closure therapy techniques, followed by early abdominal fascia closure [133,134]. To be considered optimum, a TAC approach must prevent evisceration; permit the evacuation of fluids; and prevent fascial retraction, the development of entero-atmospheric fistulas, and the loss of abdominal wall compliance [55]. The surgeon must also keep in mind that a laparostomy carries its own morbidity, including the possibility of bacterial colonization, enteric fistula, and muscle retraction. Patients who benefit from decompressive laparotomy require at least one re-exploration of the abdominal cavity before the definitive closure of the abdominal wall. For those in whom closure is not possible after the first re-exploration, one can opt for different techniques, such as fascial extensions, component separation techniques, or temporary mesh followed by a split-thickness skin graft [55]. It is recommended that the definitive closure and reconstruction of the abdominal wall be performed at an interval of 6–12 months after the last surgical intervention in order to allow for the resolution of the inflammatory process, and it involves procedures that include the visceral adhesiolysis, the restoration of digestive continuity, the closure of the fascia, and the reconstruction of the skin. One must note that recurrent ACS can occur following a premature attempt at closure [135].

Definitive closure of the abdomen can be performed in two ways: non-mesh-mediated closure techniques and techniques that use prosthetic material (mesh-mediated closure). Regarding the non-mesh-mediated technique, the “components separation technique” is considered the elective technique. This can be performed through an anterior or posterior approach and involves the dissection and translation of the parietal planes so that the visceral content can be covered. The procedure has good results and a recurrence rate of evisceration of 16% [136], but it also some negative aspects: the creation of a space that favors the formation of seromas and hematomas, and the occurrence of parietal ischemia due to the sectioning of the parietal perforating vessels. In the mesh-mediated technique, the use of biocompatible meshes is recommended. Also called “biological mesh bridging”, the technique involves the interposition of biocompatible mesh where the edges of the fascia cannot be brought together due to retraction. The negative aspect of the technique is represented by the possibility of the appearance of entero-atmospheric fistulas, due to the inflammatory process generated by the absence of a protective layer between the mesh and the visceral content [137].

## 7. Conclusions

In acute pancreatitis, the development of abdominal compartment syndrome exacerbates an already compromised clinical condition, consequently aggravating the prognosis. The prevention and screening of patients with acute pancreatitis by using intravesicular intra-abdominal pressure measurement for early diagnosis of ACS, are crucial. After a diagnosis of abdominal compartment syndrome has been made, immediate nonoperative action is required. If nonoperative techniques fail to significantly reduce the intra-abdominal pressure and improve organ function, surgical decompression should be performed without further delay.

## Figures and Tables

**Figure 1 diagnostics-13-00001-f001:**
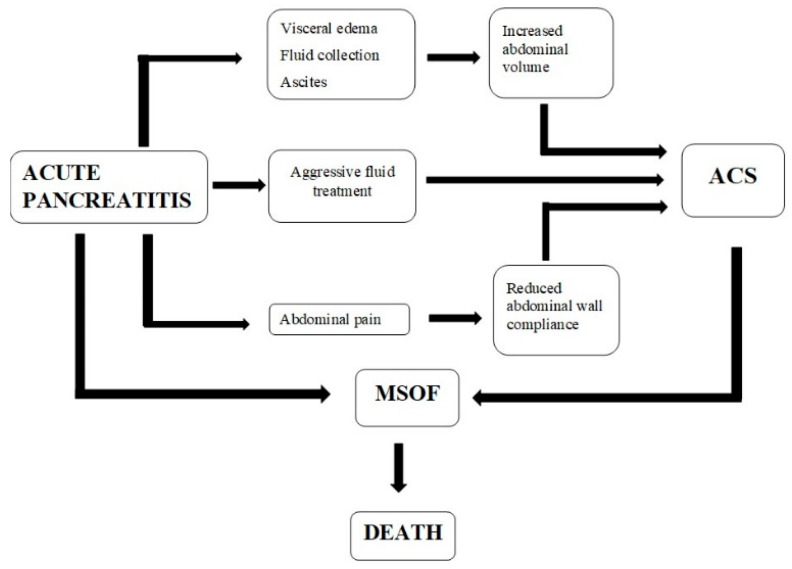
Pathophysiology of abdominal compartment syndrome in acute pancreatitis. ACS—abdominal compartment syndrome; MSOF—multiple system organ failure.

**Figure 2 diagnostics-13-00001-f002:**
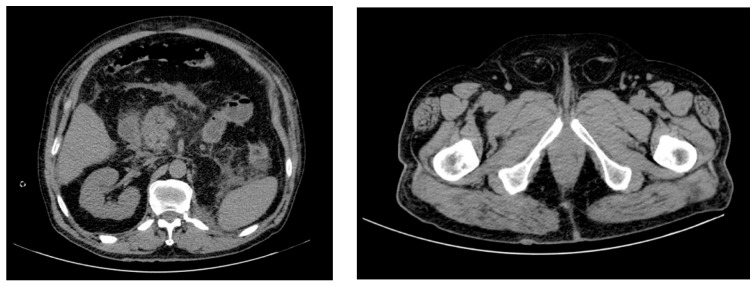
Computed tomography of the abdomen shows acute necrotizing pancreatitis complicated by abdominal compartment syndrome in a 76-year-old man; the round-belly sign and bilateral inguinal hernia can be observed.

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
