# Peer review of "Abdominal Compartment Syndrome in Acute Pancreatitis: A Narrative Review"

_diagnostics, 2022, doi:10.3390/diagnostics13010001_

Round 1

Reviewer 1 Report

The manuscript deals with an interesting and always current topic. It is well written, well organized and fluent to read. The references are adequate and recent. However, its limitation is that it represents a simple narrative of the pathology (well conducted). Its value is represented by the fact that it can represent a good starting point for future research. It can be accepted in its present form

Author Response

Dear Reviewer,

Thank you very much for your comments! We totally agree with you about the limitation of this paper. Once again, thank you very much!

Sincerely yours,

Narcis Octavian Zarnescu

Reviewer 2 Report

Dear Editor

 This is a review article for compartment syndrome of acute pancreatitis. Although the review includes many fields of acute pancreatitis, the article is not well organized, and multiple unclear points are found throughout the article. The authors discuss in some parts of the article IAH and, in some, the management of acute pancreatitis.   The following are my comments. 

#1. Line 120, please provide a reference for " 70% of patients with acute pancreatitis develop AKI"

#2. Line 158, please provide a reference for the sentence.

#3. Line 210-215 , what is your proposed method of measuring intra-abdominal pressure for the 4-stage grading?

#4. Line 219-229, as the article discusses IAH in patients with acute pancreatitis, what type of IAH occurred in pancreatitis?

#5. Line 231-249 . It is not clear IAH in acute pancreatitis is what kind of ACS ? 

#6. Line 258 , it is not clear what to do after instilling the urinary bladder with 25ml sterile saline.

#7. Line 237-234. It is not clear how to estimate IAH with POCUS when measuring only tube position and gastric content. It is better to provide illustrative figure  .

#8. Line 282-286 , the sentence is not clear to mean the use of CT for diagnosis of IAH. Please provide illustrative figures to explain. 

#9. Line 238 , why "permant" monitoring ? 

#10. Line 332, it is not clear what is "the first part " and what is "several main points"

#11. Line 390 , it is not clear about the role of antibiotics. Is it used for AP or IAH?

Author Response

Dear Reviewer,

Thank you very much for your comments and recommendations aiming to improve the quality of our manuscript! Please find bellow the answers to your questions, comments, and recommendations.

#1. Line 120, please provide a reference for " 70% of patients with acute pancreatitis develop AKI"

Reference 36: Yin J, Chen Z, Niu W, Feng L, Fan B, Zhou L, Zeng B, Zhang J, Chen H, Tong B, Tong L, Chen X. Using a multidisciplinary team for the staged management and optimally minimally invasive treatment of severe acute pancreatitis. Biosci Trends. 2021 Jul 6;15(3):180-187. doi: 10.5582/bst.2021.01075. Epub 2021 Apr 11. PMID: 33840680.

#2. Line 158, please provide a reference for the sentence.

Reference 49: Chang, Y.; Qi, X.; Li, Z.; Wang, F.; Wang, S.; Zhang, Z.; Xiao, C.; Ding, T.; Yang, C. Hepatorenal syndrome: insights into the mechanisms of intra-abdominal hypertension. Int J Clin Exp Pathol 2013, 6, 2523-2528

#3. Line 210-215 , what is your proposed method of measuring intra-abdominal pressure for the 4-stage grading?

There is no difference of measuring among the grades of the IAH.

#4. Line 219-229, as the article discusses IAH in patients with acute pancreatitis, what type of IAH occurred in pancreatitis?

Subacute: IAH that appears progressively over days, frequently found in patients with severe acute pancreatitis.

#5. Line 231-249 . It is not clear IAH in acute pancreatitis is what kind of ACS ?

According to the 2006 consensus regarding the classification of ACS, we conclude that, regarding severe acute pancreatitis, we can talk about a primary ACS.

#6. Line 258 , it is not clear what to do after instilling the urinary bladder with 25ml sterile saline.

When measuring IAP intermittently, WSACS (https://www.wsacs.org) recommends using the bladder, with a maximum instillation volume of 25 mL of sterile saline, followed by measurement at end-expiration in the complete supine position after ensuring that abdominal muscle contractions are absent and with the transducer zeroed at the level of the mid-axillary line

#7. Line 237-234. It is not clear how to estimate IAH with POCUS when measuring only tube position and gastric content. It is better to provide illustrative figure.

We agree. Unfortunately, we cannot provide this original figure at this time.

#8. Line 282-286 , the sentence is not clear to mean the use of CT for diagnosis of IAH. Please provide illustrative figures to explain.

We have inserted a new figure (Figure 2) with images of the CT scan.

#9. Line 238 , why "permanent" monitoring ?

The sense of “permanent” (line 328) was “constant screening of these patients”

#10. Line 332, it is not clear what is "the first part " and what is "several main points"

WSACS recommends a treatment algorithm that targets ACS at several key points. The first stage of treatment in the management of ACS involves non-surgical measures that can be definitive in some patients. [54,93].

#11. Line 390 , it is not clear about the role of antibiotics. Is it used for AP or IAH?

However, patients who progress to IAH are at risk for developing sepsis, necessitating the administration of antibiotics [8,86]

In addition, we have requested assistance with the English language aspects from native speakers (all from the US).

Thank you very much for your recommendations!

Sincerely yours,

Narcis Octavian Zarnescu

Reviewer 3 Report

Thanks for the chance to review this comprehensive narrative review of the pathophysiology and management of abdominal compartment syndrome in acute pancreatitis.

The authors have done an excellent job of constructing a comprehensive summary of the literature and the study should be published.

A few suggested minor improvements follow:

1. IAH is not defined until the diagnosis section. Defining the difference between ACS and IAH should be done in the Introduction to enhance reader understanding.

2. Multiple classification systems of ACS (Pahtophysiology lines 208-249). I don't think these add much to the paper and could be omitted. If included, it should be explained how these aid management.

3 Line 409 Management. It is unclear what is meant by "oral nutrition should be discontinued" - is this referring to all patients with aucte pancreatitis or only those with ACS. If those with ACS, do the authors mean enteral nutrition should be discontinued, given the recommendation for TPN? 

4. Minor English improvements :

Abstract line 13 remove "a" before "organ dysfunction"

Abstract line 13 change "in" to "on"

Abstract line 14 "resulting in significant multiple organ failure has been associated with" change to "results in significant multiple organ failures and is associated with"

Abstract line 20 change "in selected cases" to "where possible"

Introduction line 33 - suggest defining IAH here.

Introduction line 34-5 change  " a new one" to "new"

Introduction line 35 change "causing further significant" to "further increasing" 

Introduction line 42-3 remove "even though its presentation is frequently common"

Introduction line 44 change "are essential for early diagnosis and can be used as a guide in its" to "is essential for early diagnosis and underpins"

Pathophysiology line 49 change "partially" to "mostly"; change "and" to "but"

Pathophysiology line 50: Change "and has limited compliance" to ". It thus has limited compliance."

Pathophysiology line 51-52 remove "at some point", add "volume of" before "abdominal contents"

Pathophysiology line 76 change "by" to "due to"

Pathophysiology line 80 change "would build" to "builds"

Pathophysiology line 82 change "play" to "have"; change "in" to "on"

Similar edits are required throughout the remaining paper.

Overall the review is comprehensive and worthy of publication.

Author Response

Dear Reviewer,

Thank you very much for your comments and recommendations aiming to improve the quality of our manuscript!

Please find bellow the answers to your questions, comments, and recommendations.

  1. IAH is not defined until the diagnosis section. Defining the difference between ACS and IAH should be done in the Introduction to enhance reader understanding.

             IAH is defined as intra-abdominal pressure values over 12 mmHg, obtained after two measurements at intervals of 1-6 hours [7]

  1. Multiple classification systems of ACS (Pahtophysiology lines 208-249). I don't think these add much to the paper and could be omitted. If included, it should be explained how these aid management.

Grades I and II can be treated conservatively, but grades III and IV require surgery.

  1. Line 409 Management. It is unclear what is meant by "oral nutrition should be discontinued" - is this referring to all patients with aucte pancreatitis or only those with ACS. If those with ACS, do the authors mean enteral nutrition should be discontinued, given the recommendation for TPN?

              Oral nutrition should be discontinued, and total parenteral nutrition should be commenced in patients with manifest abdominal compartment syndrome.

  1. Minor English improvements :

Abstract line 13 remove "a" before "organ dysfunction"

Abstract line 13 change "in" to "on"

Abstract line 14 "resulting in significant multiple organ failure has been associated with" change to "results in significant multiple organ failures and is associated with"

Abstract line 20 change "in selected cases" to "where possible"

Introduction line 33 - suggest defining IAH here.

Introduction line 34-5 change  " a new one" to "new"

Introduction line 35 change "causing further significant" to "further increasing"

Introduction line 42-3 remove "even though its presentation is frequently common"

Introduction line 44 change "are essential for early diagnosis and can be used as a guide in its" to "is essential for early diagnosis and underpins"

Pathophysiology line 49 change "partially" to "mostly"; change "and" to "but"

Pathophysiology line 50: Change "and has limited compliance" to ". It thus has limited compliance."

Pathophysiology line 51-52 remove "at some point", add "volume of" before "abdominal contents"

Pathophysiology line 76 change "by" to "due to"

Pathophysiology line 80 change "would build" to "builds"

Pathophysiology line 82 change "play" to "have"; change "in" to "on"

Similar edits are required throughout the remaining paper.

All these suggestions were addressed. In addition, we have requested assistance with the English language aspects from native speakers (all from the US).

Thank you very much for your recommendations!

Sincerely yours,

Narcis Octavian Zarnescu

Round 2

Reviewer 2 Report

The authors response to my questions raised and I have no more co,mments